# Study on ring-road incident duration based on latent class accelerated hazard model

Qiangru Shen[1], Xun Xie[2], Gen Li[ID][2]*, Lan Wu[2], Le Zhao[2], Zhen Yang[2]

**1** School of Transportation and Civil Engineering, Nantong University, Nantong, Jiangsu, China, **2** College of Automobile and Traffic Engineering, Nanjing Forestry University, Nanjing, Jiangsu, China

* ligen@njfu.edu.cn

**Data Availability Statement:** All data are fully available without restriction. The data used in this study can be accessed at Figshare by using the following link: https://doi.org/10.6084/m9.figshare.26094055.

## Abstract

Accurately estimating the duration of freeway incidents can enhance emergency management practices and reduce the likelihood of secondary incidents. To investigate the mechanisms through which key factors influence incident duration, this study sorted out the characteristics and variables of the incident duration on a special freeway in Zhejiang Province, that is, the ring road, and developed a latent class accelerated hazard model. Heterogeneity was incorporated into the model. Three distributions (Weibull, Log-normal, and Log-logistic) were compared, and the Log-logistic distribution exhibited superior performance. The analysis revealed two distinct latent classes: Latent Class 1 and Class 2, had class membership probability of 0.53 and 0.47, respectively, with a total of 11 variables being statistically significant at the 0.05 significance level. It is worth noting that, some neglected explanatory variables are discussed in depth in this study. For example, the mechanism of which specific lane is closed has an impact on the incident duration, rather than a general discussion of the number of lane closures. Furthermore, the way in which the driver involved in the incident reports to the police has a significant impact on the duration of incidents. Notably, potential heterogeneity and its influencing mechanism are captured in the model. Additionally, by predicting class membership using posterior probabilities, it was determined that most data points were more likely to belong to Class 1, and the incident duration primarily ranged between 0 and 60 minutes. These findings are helpful to reduce the duration of incidents on ring-roads and freeways in China, and provide theoretical support for the formulation of freeway incident management and treatment policies.

## Introduction

To mitigate congestion, delays, and the occurrence of secondary incidents on freeways, it is imperative to examine the overall duration of incidents and establish a theoretical framework for enhancing the level of freeway emergency management. As defined in the Highway Capacity Manual, the total incident duration encompasses detection time, response time, clearance time, and recovery time [1].

To further investigate incident duration, various statistical methods have been employed to explore the relationship between incident duration and its influencing factors. Among these,

**Funding:** The author(s) received no specific funding for this work.

**Competing interests:** The authors have declared that no competing interests exist.

linear regression is an early method used in analyzing incident duration. Giuliano conducted a linear regression analysis on incident data from a major freeway in Los Angeles, California, revealing variations in incident duration based on incident type, lane closures, and time of day [2]. Khattak et al. utilized truncated regression models to estimate the factors influencing incident duration and improved the model fit by introducing additional variables [3]. Ding et al. proposed a switching regression model and a binary probit model to examine the factors affecting incident duration [4]. However, regression models only assume a linear relationship between incident processing time and the influencing factors. In contrast, hazard-based duration models not only consider the processing time but also the relationship between duration and the probability of ending within the next short time interval [5]. Breslow initially introduced the proportional hazards (PH) model, which formed the basis for the development of various hazard-based duration models, including the accelerated failure time (AFT) model and the Cox proportional hazards regression model [6–10]. Among them, AFT model has been widely used because of its good applicability to data with trailing [11–13]. Similarly, Alkaabi et al. demonstrated the significant impact of incident characteristics on incident duration using a Weibull hazard duration model without gamma heterogeneity. The factors considered included month, location, weather conditions, incident type, number of casualties, and number of vehicles involved [14]. Tavassoli et al. compared incident duration models by considering fixed and random parameters under different distributions. Their study suggested that the Weibull AFT model with random parameters is suitable for simulating collision-induced incident durations, while the Weibull model with gamma heterogeneity is suitable for modeling incident durations of stationary vehicles [15]. Li et al. compared generalized gamma, Weibull, and Log-logistic distributions with fixed and random parameters to determine the most appropriate model for analyzing and predicting incident processing time [16]. Most of the aforementioned studies have relied on data from foreign freeways and have identified similar significant factors that influence incident duration, including vehicles involved, incident locations, peak hours, and weather conditions. However, these studies have overlooked potential differences in traffic composition and operational management of freeways in developing countries like China. Consequently, there is an urgent need for localized research on traffic safety theories. Furthermore, the rapid development of information technology and vehicle intelligence in recent years has resulted in more complex changes in freeway operations and traffic composition. The factors previously examined in studies may have certain limitations, underscoring the necessity for further in-depth research on the duration of traffic incidents on freeways.

Additionally, Ozbay and Noyan employed a Bayesian network model to estimate incident clearance time [17]. Ghosh et al. compared hazard-based duration models using the generalized F-distribution and determined that the generalized F-distribution was the most suitable for analyzing incident duration data [18]. Zou et al. analyzed incident duration using a finite mixture model to explore the factors influencing duration. The finite mixture model yielded improved estimations and predictions of freeway incident duration compared to the AFT model [19]. Subsequent research by Zou et al. indicated that quantile regression outperformed the Log-logistic AFT and Cox proportional hazards duration models in analyzing factors affecting freeway incident duration and incident clearance time [20]. In a recent study, Tang et al. compared four statistical methods and four machine learning methods using incident duration data and found that the random parameter hazard duration model exhibited the best fit. Notably, statistical methods demonstrated better capacity to address latent defects in the data than machine learning methods [21]. Interestingly, Bai et al. proposed a combination of the Kaplan-Meier (K-M) model and random survival forest (RSF) model to predict the duration of geo-hazard incidents, providing reference significance for further research on incident

duration using machine learning methods [22]. While the aforementioned studies have utilized statistical models or machine learning methods to analyze incident duration to some extent, to the best of our knowledge, few studies in this field have employed a latent class accelerated hazard model for a comprehensive analysis. We found that the latent class accelerated hazard model effectively revealed potential heterogeneity in incident duration data and enhanced our understanding of the trailing phenomenon within the data.

The objective of this research is to investigate the duration of freeway incidents specifically in the context of a ring-road, utilizing the latent class accelerated hazard model. Furthermore, due to variations in freeway management policies that are influenced by national characteristics, the findings from prior studies on foreign freeway incident duration have limited applicability to domestic freeways. Thus, in order to empirically analyze the subject matter of this paper, we collected and analyzed one-year incident duration data from a ring-road in Zhejiang Province, China. As a special freeway in China, the ring-road is mainly used to relieve the traffic pressure in urban areas, divert vehicles, and build a three-dimensional freeway network by forming an interchange with various high-grade freeways leading to the main urban area. The ring-road has some characteristics besides the general freeway, such as weak tidal property, many entrances and exits, complex sections, etc. Building upon previous research experience and considering the unique influencing factors of Chinese freeways, we examined the mechanisms by more different variables which impact incident duration and analyzed the effects of unobserved heterogeneity in the latent classes in depth. The findings of this study can contribute to the localized research on incident duration of special freeways. Moreover, they can serve as a reference for local authorities in formulating and improving relevant policies, optimizing freeway monitoring systems, and allocating resources more effectively. This, to some extent, helps in reducing the harm and losses caused by freeway incidents.

The paper is structured into four sections. The Materials and Methods section presents the freeway incident duration data used in this study and outlines the latent class accelerated hazard model. In Results section, we discuss the selection of the optimal number of classes in the model and present the results of model estimation. The Discussion section compares and discusses the similarities and differences between the findings of this study and previous research. Finally, the Conclusion section draws conclusions from our findings and provides insightful suggestions for future research.

## Materials and methods

### Data

This study utilized incident data from a ring-road in Zhejiang Province in 2021, which was provided by relevant authorities. On-site incident response personnel recorded specific incident information. The studied section of the freeway underwent expansion and reconstruction in 2012, transitioning from a four-lane, two-way road to an eight-lane, two-way road. The dataset consisted of a substantial amount of real incident data that naturally occurred. The average duration of incidents in this dataset was 30.13 minutes, with a standard deviation of 38.55 minutes. The maximum incident duration recorded was 7 hours and 32 minutes, while the minimum was 1 minute. Moreover, 22 percent of the incidents resulted from vehicles colliding with fixed objects, while only 2 percent involved vehicle tipping over or catching fire. Additionally, 28 percent of the incidents transpired during the nighttime, and 34 percent of incidents involved individuals using mobile phone alarms. Approximately 40 percent of the incidents required the closure of the fourth lane, and 19 percent of incidents occurred during rainy weather.

**Ethical approval and consent to participate.** The data used in the current investigation was collected previously by transportation administration staff for incident management and subsequent incident studies. Because the data used in this study does not involve the privacy of individuals and human research participants, no relevant approvals and consents are required. In data collection, there was no written or verbal consent from participants. The reason was the investigators did NOT get participants rather, secondary data were obtained in the chart of incident records. The data was de-identified by the investigators.

The overall data processing procedure of this paper is presented in Fig 1. Using the latent class accelerated hazard model as a basis, we transformed a total of 42 potential explanatory variables into dummy variables and grouped them into incident characteristics (e.g., incident type, incident location, etc.), temporal characteristics (e.g., nighttime, peak hours, etc.), environmental characteristics (e.g., sunny, rainy, etc.), traffic characteristics (e.g., lane closure, etc.), and operational characteristics (e.g., incident reporting method, etc.) [23]. We employed stepwise regression to eliminate variables with weak effects and utilized the Akaike

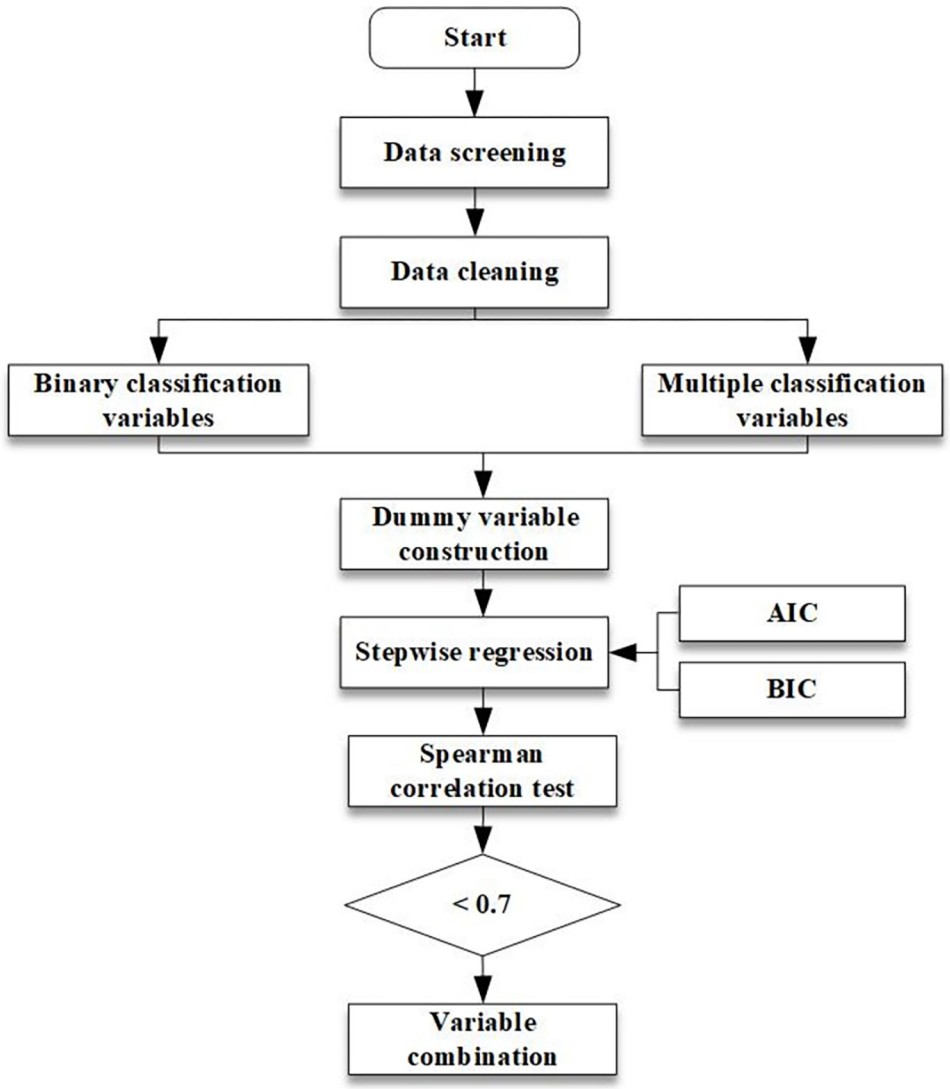

**Fig 1. Flowchart of data processing.**

**Table 1.  Description of each variable in the optimal variable combination.**

| Variables | Variable description |
|---|---|
| **Incident characteristics** | |
| Objects | The incident was caused by a vehicle striking fixed objects. |
| Scraping | The incident was caused by vehicle scraping. |
| Rollover | Vehicle tipping occurred during the incident. |
| Fire | A vehicle fire broke out during the incident. |
| HV | Heavy vehicles were involved in the incident. |
| Serious | The event is defined as a serious event. |
| Multivehicle | Multiple vehicles were involved in the incident. |
| **Temporal characteristics** | |
| Night | The incident was happened at night. |
| **Environmental characteristics** | |
| Rainy | The incident happened on a rainy day. |
| **Traffic characteristics** | |
| Lane4 closure | The fourth lane will be closed after the incident. |
| **Operational characteristics** | |
| Mobile phone alarm | The individuals involved choose to report the incident through their mobile phones as a means to seek assistance or rescue. |

Information Criterion (AIC) and Bayesian Information Criterion (BIC) as evaluation criteria. The resulting optimal combination of variables from the stepwise regression analysis demonstrated the significance of factors such as collisions with fixed objects, scrapes, rollovers, fires, nighttime incidents, mobile phone alarms, lane4 closures, heavy vehicles, incident severity, multi-vehicle incidents, and rainy weather conditions. Further details on the variables in this combination are provided in Table 1. The existing research focuses on the number of lane closures [8, 23]. For the ring-road, which lane to close is more meaningful for the application of research and the handling of incidents.

To further verify whether there is multicollinearity and tight correlation in the variable combinations after stepwise regression, the Spearman correlation test was conducted to assess the relationship between the aforementioned variables. The results revealed that the absolute values of the correlation coefficients were less than 0.7, as displayed in Table 2. This shows that

**Table 2.  Correlation matrix of optimal variable combination.**

| | Objects | Scraping | Rollover | Fire | Night | Phone | Lane4 | HV | Serious | Multivehicle | Rainy |
|---|---|---|---|---|---|---|---|---|---|---|---|
| Objects | 1.000 | | | | | | | | | | |
| Scraping | -0.124 | 1.000 | | | | | | | | | |
| Rollover | -0.068 | -0.030 | 1.000 | | | | | | | | |
| Fire | -0.046 | -0.020 | -0.011 | 1.000 | | | | | | | |
| Night | 0.197 | -0.042 | 0.070 | 0.066 | 1.000 | | | | | | |
| Phone | 0.328 | -0.114 | 0.035 | 0.007 | 0.230 | 1.000 | | | | | |
| Lane4 | 0.223 | -0.042 | 0.004 | 0.063 | 0.060 | 0.176 | 1.000 | | | | |
| HV | -0.088 | 0.067 | 0.121 | 0.103 | 0.106 | 0.101 | 0.170 | 1.000 | | | |
| Serious | -0.087 | -0.052 | 0.007 | -0.019 | 0.094 | -0.003 | 0.016 | 0.157 | 1.000 | | |
| Multivehicle | -0.171 | -0.076 | -0.054 | -0.037 | -0.054 | -0.099 | -0.066 | -0.089 | 0.050 | 1.000 | |
| Rainy | 0.063 | -0.047 | 0.015 | -0.014 | 0.056 | 0.044 | 0.001 | 0.013 | -0.052 | -0.014 | 1.000 |

there is no tight correlation in the optimal combination of variables obtained by stepwise regression, further excluding the additive effects of other factors.

## Methodology

**Description of permission to work.**   The data used in the study was provided by the relevant freeway management departments and recorded by relevant staff on site. This study has obtained the informed consent of the data provider.

In order to make up for the disadvantage that the proportional hazard models assume that the influence of explanatory variables on the hazard function is proportional, and there is no direct relationship between the covariates and the target variable, the accelerated hazard models are proposed and further applied to reliability theory and industrial experiments. At present, acceleration hazard models based on random parameters are widely used in the study of incident duration because of their ability to reveal unobserved heterogeneity. Another way to incorporate heterogeneity into accelerated hazard models is the latent class model. Unlike random parameter models, latent class models assume that the observed data comes from a population with several subpopulations or components that can naturally be divided into several latent classes. In this study, the total duration of the incident is used as the failure time in the accelerated hazard model.

In the latent class approach, the incident duration data is grouped into segments and the parameters for each segment are estimated to capture the differences between the effects of the parameters on the incident duration. In this method, assuming that there are different homogeneous latent classes and that the data of an incident duration belongs to latent class $k$, the latent class accelerated hazard model can be expressed as:

$$Y = \sum_{k=1}^{K} \omega_k (\boldsymbol{\beta}_k \mathbf{X} + \boldsymbol{\varepsilon}_k) \tag{1}$$

Where $Y$ is the logarithm of incident duration; $\mathbf{X}$ is the covariate vector; $\omega_k$ is the proportion of latent class $k$; $\beta_k$ is the unknown parameter vector for each latent class $k$; $\varepsilon_k$ is the error term of class $k$.

The latent class model assumes that the observations are distributed heterogeneously with a discrete distribution within the population, and the distribution cannot be directly observed. In general, the discrete distribution is specified in multinomial logit form.

$$\begin{cases} \omega_{ik} = \dfrac{\exp(\nu_k + \boldsymbol{\theta}_k \boldsymbol{\xi}_i)}{\sum\limits_{k=1}^{K} \exp(\nu_k + \boldsymbol{\theta}_k \boldsymbol{\xi}_i)} \\[2ex] \sum\limits_{k=1}^{K} \omega_k = 1 \\[1ex] \nu_k = 0 \\[0.5ex] \boldsymbol{\theta}_k = 0 \end{cases} \tag{2}$$

Where $\nu_k$ is the intercept of class $k$; $\boldsymbol{\theta}_k$ is the vector of the correlation coefficient; $\boldsymbol{\xi}_i$ is the carrier of individual characteristics; $i$ is the explanatory variable corresponding to the incident

duration. If no individual characteristics are provided, the formula can be expressed as:

$$\begin{cases} \omega_{ik} = \dfrac{\exp(v_k)}{\displaystyle\sum_{k=1}^{K}\exp(v_k)} \\ \displaystyle\sum_{k=1}^{K}\omega_k = 1 \\ v_k = 0 \end{cases} \tag{3}$$

The log-likelihood of the latent class accelerated hazard model can be expressed as:

$$\ln L = \sum_{i}^{N}\ln L_i = \sum_{i}^{N}\ln\left(\sum_{k=1}^{K}\omega_{ik}L_{i|class=k}\right) \tag{4}$$

Based on the estimation of the current parameters, the posterior probability that the explanatory variable $i$ belongs to class $k$ can be calculated as:

$$\hat{\omega}_{ik} = \dfrac{\omega_{ik}L_{i|class=k}}{\displaystyle\sum_{k=1}^{K}\omega_{ik}L_{i|class=k}} \tag{5}$$

Usually, expectation-maximization (EM) algorithm is applied to efficiently estimate latent class model in three steps:

(1) Initialize the seeds of all parameters using K-means clustering or other methods;

(2) Step E: Estimate the posterior component probability of each observation by Eq (5), and derive the mixing proportions as follows:

$$\hat{\omega}_k = \frac{1}{N}\sum_{i=1}^{N}\hat{\omega}_{ik} \tag{6}$$

(3) Step M: According to Eq (6), the posterior probability is used as the weight, and the log-likelihood of each component is maximized respectively to obtain a new estimates of the parameters.

The EM algorithm can alternate between desired and maximized steps until the likelihood improvement falls below a prespecified threshold or reaches a maximum number of iterations. However, the number of classes cannot be directly estimated, so Bayesian information criteria (BIC) is used to determine the number of $K$.

$$BIC = -2LL + \gamma\log(N) \tag{7}$$

Where LL is the log-likelihood value; $\gamma$ is the number of free parameters to be estimated; $N$ is the number of observations in the data. In this paper, the components of latent class accelerated hazard model are set from 1 to 6 for comparison, and the best model components are selected according to BIC.

## Results

### Selection of classes

Using incident data from a specific ring-road in Zhejiang Province, the optimal number of classes for the latent class model was determined based on the methodology described in the Materials and Methods section of this paper. Fig 2 displays the BIC values of the latent class

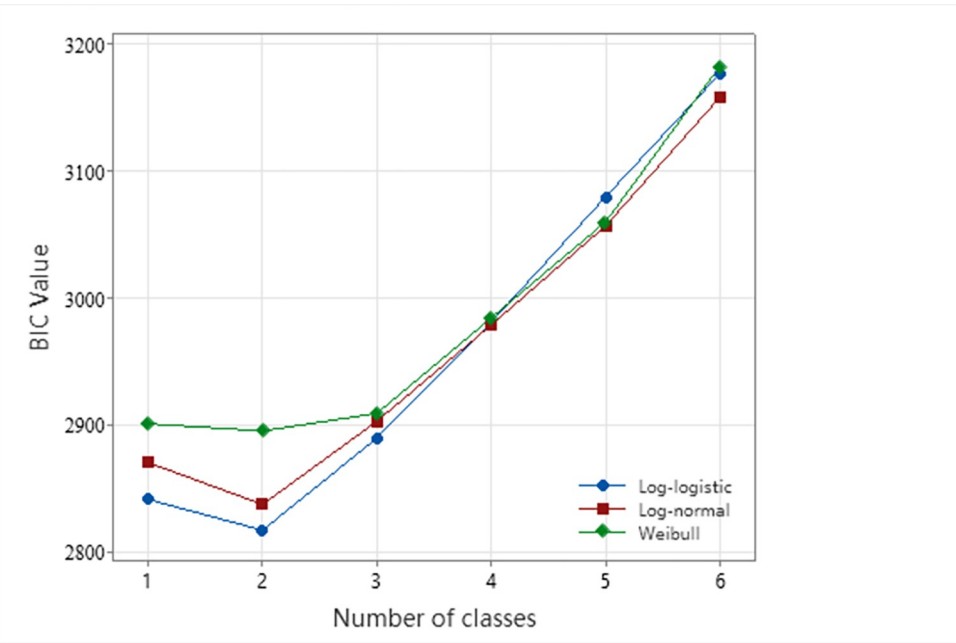

**Fig 2. Selection of the optimal number of classes under different distributions.**

model as the number of classes ranges from 1 to 6, indicating the model's optimal performance when there are 2 classes under the Log-logistic distribution. Additionally, Table 3 presents the AIC, BIC, and log-likelihood values for the 2-class model under various distributions. The results consistently favor the Log-logistic distribution as the best choice across all indicators. Therefore, the Log-logistic model was selected for parameter estimation and subsequent analysis.

## Results

In order to account for potential heterogeneity among variables, this paper employs a heterogeneity model, specifically the latent class model, to determine the proportion of data influenced positively and negatively. The estimation results of the model are presented in Table 4, where positive parameter estimates indicate a longer incident duration. The latent class model identifies two distinct classes: latent class 1 with a membership probability of 0.532 and latent class 2 with a membership probability of 0.468. These classes are found to be statistically significant for the incident dataset, and no further improvement in model fit is observed with more than two classes.

To enhance our understanding of the variables' impact on incident duration, Table 4 additionally presents the percentage change in incident duration attributed to each significant variable in the latent class accelerated hazard model, including some explanatory variables unique

**Table 3. Performance comparison of latent class model distribution form.**

|  | Incident duration | | |
| --- | --- | --- | --- |
|  | **Loglogistic** | **Lognormal** | **Weibull** |
| AIC | 2683.2 | 2703.6 | 2761.3 |
| BIC | 2817.3 | 2837.6 | 2895.4 |
| LL | -1314.6 | -1324.8 | -1353.7 |

**Table 4. Parameter estimation results of latent class model.**

| Variables | Latent Class 1 | | | Latent Class 2 | | |
|---|---|---|---|---|---|---|
| | Coefficient | Z | Percentage change (%) | Coefficient | Z | Percentage change (%) |
| Constant | 3.100*** | 31.58 | / | 1.677*** | 9.92 | / |
| **Incident characteristics** | | | | | | |
| Objects | 0.190* | 1.95 | 20.9 | 0.409** | 2.40 | 50.5 |
| Scraping | -0.890*** | -5.73 | -58.9 | -0.826*** | -3.91 | -56.2 |
| Rollover | 0.635*** | 2.77 | 88.7 | 2.579*** | 6.09 | 1218.4 |
| Fire | 0.600** | 2.24 | 82.2 | 0.937 | 1.59 | 155.2 |
| Heavy vehicle | 0.363*** | 5.26 | 43.8 | 0.191 | 1.51 | 21.0 |
| Serious | 0.459*** | 3.80 | 58.2 | 1.939*** | 6.14 | 595.2 |
| Multi-vehicle | 0.394*** | 4.88 | 48.3 | 0.444*** | 3.18 | 55.9 |
| **Temporal characteristics** | | | | | | |
| Nighttime | 0.061 | 0.82 | 6.3 | 0.353*** | 2.94 | 42.3 |
| **Environmental characteristics** | | | | | | |
| Rain | 0.167** | 2.35 | 18.2 | 0.095 | 0.68 | 10.0 |
| **Traffic characteristics** | | | | | | |
| Lane4 closure | 0.039 | 0.52 | 4.0 | 0.559*** | 4.97 | 74.9 |
| **Operational characteristics** | | | | | | |
| Mobile phone alarm | -0.193** | -2.09 | -17.6 | 0.587*** | 3.21 | 79.9 |
| **Model structure parameters** | | | | | | |
| Sigma | 0.295*** | 9.34 | / | 0.501*** | 17.27 | / |
| Latent class probability | 0.532 *** | 6.59 | / | 0.468 *** | 5.79 | / |
| Log-likelihood | -1314.6 | | | | | |

Note

*, ** and *** was statistically significant at 0.1, 0.05, and 0.01 levels, respectively.

to the ring-road that were not discussed in previous studies. Based on these findings, the explanatory variables are categorized into incident characteristics, temporal characteristics, environmental characteristics, traffic characteristics, and operational characteristics. This paper further explores and discusses these variables based on their respective categories.

(1) Incident characteristics

Table 4 demonstrates that the scraping indicator in incident characteristics results in a 58.9% reduction in incident duration for latent class 1 and a 56.2% reduction in latent class 2. This greater reduction in latent class 1 suggests that scrape incidents have less impact and are more manageable. The impact of colliding fixed objects, heavy vehicles, and multi-vehicle incidents on incident duration is comparable, with a positive effect on increasing duration. Notably, the rollover indicator shows a significant increase in incident duration in latent class 2, which may indicate more severe incidents compared to latent class 1. Similarly, the serious incidents indicator exhibits a 595.2% increase in incident duration in latent class 2, while latent class 1 shows a 58.2% increase. Fire incidents also result in longer incident duration in both latent classes, despite the indicator not being significant in latent class 2.

(2) Temporal characteristics

Research has consistently demonstrated that traffic incidents occurring at night tend to be more severe [5]. The findings from the latent class model analysis reveal that in latent class 2, nighttime incidents have significantly longer durations compared to latent class 1, with an increase of 42.3% in incident duration. This observation suggests that incidents belonging to latent class 2 exhibit a greater degree of severity than those in latent class 1. Moreover, reduced

visibility during nighttime conditions may disrupt emergency response and handling procedures, potentially contributing to longer incident durations.

(3) Environmental characteristics

Table 4 indicates that incidents occurring during rainy weather are associated with an increase in incident duration (18.2% in latent class 1 and 10.0% in latent class 2). This can be attributed to reduced visibility on rainy days, which affects emergency response personnel, consequently prolonging the duration of incidents.

(4) Traffic characteristics

In this study, in order to better analyze the specific effect of lane closure on the duration of the incident, the specific location of the lane closure is selected, that is, which lane is closed, rather than the number of lanes closed [23] in general. The results showed that the closure of the fourth lane shows positive impact on increasing incident duration in both latent classes. In latent class 2, incident duration increases by 79.9% as a result of lane 4 closure, whereas in latent class 1, it only increases by 4.0%. This disparity can be attributed to differences in management policies and laws and regulations of ring-roads, where trucks are typically required to travel on the right-hand side, resulting in a higher concentration of trucks on the fourth lane. As a result, incidents that occur in the fourth lane tend to be more severe and have longer durations. Thus, it is important to acknowledge that research findings from foreign countries may not be directly applicable to developing countries like China due to regulatory differences in managing freeways.

(5) Operational characteristics

Due to the rapid development of information technology in recent years, the composition of freeway operations has undergone more complex changes, and the way drivers report an incident to the police, that is, via mobile phone reporting, has a significant impact on the duration of incidents. The data presented in Table 4 reveals discrepancies in the performance of the mobile phone alarm indicator across the two latent classes. Within latent class 1, which represents 53.2% of incidents, the mobile phone alarm has a negative sign, signifying its contribution to reducing incident duration. Conversely, in latent class 2, accounting for 46.8% of incidents, the variable is associated with an increase in incident duration, suggesting some heterogeneity in the variable across the two latent classes. It is important to note that since there is a larger amount of data in latent class 1, it suggests that utilizing the mobile phone alarm has a positive impact on decreasing incident duration. Further investigation demonstrates that incident duration decreases by 17.6% in latent class 1, but experiences a significant increase of 79.9% in latent class 2. This disparity may be attributed to the higher complexity of the incident scene in latent class 2 and unclear information provided by the caller using the mobile phone alarm, which poses greater challenges for emergency response.

This paper utilizes the posterior probability to predict class membership, revealing the distribution of incident duration across two latent classes, as depicted in Fig 3. The analysis indicates that the majority of incident duration data are more likely to belong to class 1. Furthermore, class 1 comprises incident durations that tend to fall within the middle or shorter term, concentrating between 0 and 60 minutes. This finding suggests that the latent class accelerated hazard model more effectively identifies exceptional incident duration data and categorizes it accordingly, providing an improved understanding of the tail phenomena observed in the incident duration distribution.

## Discussion

Previous research [4, 20, 24] has indicated the significant impact of the number of vehicles involved in incidents on their duration, with multi-vehicle incidents generally having longer

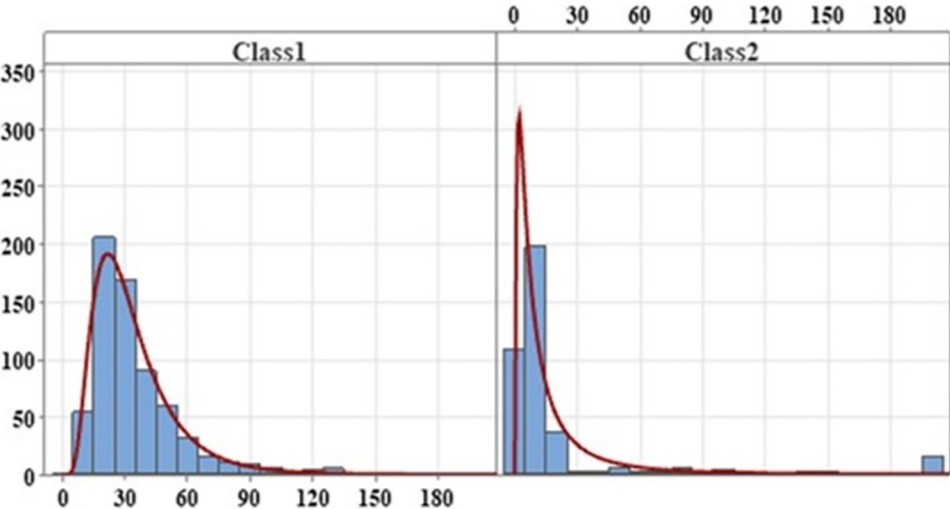

**Fig 3. Distribution of incident duration for the two classes.**

durations compared to single-vehicle incidents. Additionally, night-time and rainy conditions have been identified as key factors affecting incident duration [15, 18, 23]. These factors can interfere with incident response and handling, leading to extended durations. This study yielded similar findings, emphasizing the factors that substantially increase the duration of incidents and warrant greater attention. The latent class accelerated hazard model results highlight the significance of rollover incidents, fires, and serious incidents, calling for their prioritization. Thus, it is crucial for local authorities to implement appropriate measures, including considering the location and equipment of rescue centers, as well as ensuring the availability of vehicles such as fire trucks and tow trucks. Moreover, longer incident durations are often associated with collisions with fixed objects and multi-vehicle incidents, underlining the need for strengthening the freeway network's systems. This includes accurate assessment of heavy vehicles and incident scenes, as well as enhancing the efficiency of medical personnel to minimize harm and losses resulting from freeway incidents. Additionally, nighttime and rainy conditions are identified as important factors contributing to increased incident duration. These conditions affect response times for incident clearance by relevant departments and lead to reduced visibility. To expedite the response time for such challenging incidents, resources within the freeway domain should be reallocated towards the identification and reporting of incident scenes. Local management authorities should also consider equipping resources, such as lighting and protective facilities, to enhance the efficiency of incident clearance procedures. Of course, more specific descriptions of explanatory variables often lead to more valuable applications of research findings. Compared with discussing the number of lane closures, the specific location of lane closures has a higher guiding value for the formulation of policies to reduce the duration of incidents. In China, since heavy vehicles are mostly distributed in the fourth lane, closing the fourth lane will lead to an increase in the duration of the incident. Local authorities could consider shifting some of their resources from responding to incidents in other lanes to the fourth lane and strengthening the management of heavy vehicles in the hope of reducing the duration of incidents.

Previous studies [16, 23, 25] have indicated the significant impact of peak hours on incident duration. However, contrary to expectations, the results of this study indicate that the indicator of peak hours in the temporal characteristics is not statistically significant in the model fit. This disparity can be attributed to the fact that the object in this study is a special kind of

freeway in China: the ring-road. Compared to general freeways, the ring-road is mainly used to relieve the traffic pressure in urban areas rather than for commuting, so tidal phenomena are weaker. In addition, another possibility is the notable differences in traffic composition and operational management between domestic and foreign freeways. Foreign freeways are often characterized by commuting purposes and more prominent tidal phenomena compared to Chinese freeways. These disparities highlight the distinct traffic safety characteristics between domestic and foreign freeways and caution against directly applying research conclusions based on foreign freeway incident data to the management and operation of domestic freeways. In terms of freeway emergency management, it is crucial to prioritize understanding the differences in factors influencing freeway, especially on special freeway, such as ring-road incident duration at the domestic and international levels.

The research findings reveal that the impact of reporting incidents via mobile phone varies across the two latent classes. Considering the larger amount of data in latent class 1, it can be concluded that reporting incidents via mobile phone has a positive effect on reducing incident duration. This variable appears due to the rapid development of information technology, and considering the potential effects of varying freeway management policies and practices in China, in-depth discussion of this variable fills the gap in previous studies. Local authorities can promote the use of mobile phone reporting as part of their freeway management policies to facilitate incident clearance. However, it is crucial for incident participants to provide clear and accurate descriptions of the incident scene. Moreover, local management authorities must enhance relevant technologies to precisely locate the incident scene and evaluate the situation when incident participants report incidents via mobile phone. This will facilitate the effective allocation of resources and ensure efficient rescue operations.

## Conclusions

In this study, the duration data of incidents on a particular ring-road in Zhejiang Province were fitted using the latent class accelerated hazard model, with heterogeneity incorporated. Three distributions (Weibull, Log-normal, and Log-logistic) were compared, and the results revealed that the Log-logistic distribution provided the best fit for the latent class accelerated hazard model.

In the latent class accelerated hazard model, two classes were determined as optimal, and most explanatory variables exhibited similar effects in both classes, also capturing the effect of potential heterogeneity. The results showed that in the ring-road, variables such as collisions with fixed objects, rollover, fire, heavy vehicles, severe incidents, multi-vehicle incidents, nighttime, rainy weather, and closure of lane 4 were found to lead to longer incident durations in both latent classes. Conversely, the scraping indicator was associated with shorter incident durations in both classes. Based on these results, several recommendations can be made for freeway management and maintenance departments to reduce the occurrence of secondary incidents caused by long incident durations.

More and more specific and evolving variables should be taken into account. Compared with the impact of the number of closed lanes on the duration of incidents, the conclusion that closing the fourth lane will lead to an increase in the incident duration has more guiding value for the formulation of policies and the handling of incidents. Additionally, the findings confirm that freeway conditions in developing countries, such as ring-road in China, differ from previous research findings. Peak hours have a relatively diminished impact on incident duration in Chinese freeways, and the use of mobile phones for incident reporting is prominent. As such, incorporating mobile phone usage into management policies and leveraging relevant technologies are essential for improving incident duration accuracy and enhancing the

efficiency of incident handling operations. The recommendations presented based on the research findings of this study can serve as valuable references for local authorities aiming to improve relevant policies and management systems. Implementation of these recommendations is expected to effectively mitigate the harm and losses resulting from freeway incidents.

Furthermore, these findings highlight the effectiveness of the latent class accelerated hazard model in capturing the heterogeneity within incident duration data. It is also found that the model can more naturally identify anomalies in the incident duration data and better explain the trailing phenomenon in the data.

Nonetheless, several limitations should be acknowledged in this study. Firstly, only one year of freeway incident duration data was examined, potentially overlooking the effects of time on incident duration. Additionally, driver attribute variables were not included in the incident duration dataset. For future studies, it is recommended to incorporate comprehensive datasets that consider time effects and diverse variables. Moreover, the transferability of the analysis results to other freeways in China should be explored to ensure the generalizability of the findings.

## Acknowledgments

The authors of this article would like to thank the Zhejiang Scientific Research Institute of Transport for making available the data.

## Author Contributions

**Methodology:** Xun Xie, Gen Li.

**Software:** Le Zhao.

**Visualization:** Lan Wu.

**Writing – original draft:** Qiangru Shen.

**Writing – review & editing:** Zhen Yang.

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
