## [Decision Letter · Decision Letter 0]

5 Jun 2024

PONE-D-24-18829Study on ring-road incident duration based on latent class accelerated hazard modelPLOS ONE

Dear Dr. Li,

Thank you for submitting your manuscript to PLOS ONE. After careful consideration, we feel that it has merit but does not fully meet PLOS ONE’s publication criteria as it currently stands. Therefore, we invite you to submit a revised version of the manuscript that addresses the points raised during the review process.

We look forward to receiving your revised manuscript.

Kind regards,

Zhihong (Arry) Yao, Ph.D.

Academic Editor

PLOS ONE

Journal Requirements:

Reviewers' comments:

Reviewer's Responses to Questions

**Comments to the Author**

1. Is the manuscript technically sound, and do the data support the conclusions?

Reviewer #1: Yes

Reviewer #2: Yes

2. Has the statistical analysis been performed appropriately and rigorously? 

Reviewer #1: Yes

Reviewer #2: Yes

3. Have the authors made all data underlying the findings in their manuscript fully available?

Reviewer #1: Yes

Reviewer #2: Yes

4. Is the manuscript presented in an intelligible fashion and written in standard English?

Reviewer #1: Yes

Reviewer #2: Yes

5. Review Comments to the Author

Reviewer #1: This manuscript explores factors that potentially influence the incident duration of a unique freeway in China (ring-road). Incident duration is critical for freeway emergency management. Overall, this paper comprehensively considers various influencing factors and captures the effects of potential heterogeneity. My feedback on the study is as follows:

The introduction of the article summarizes the contributions made in the field of incident duration research in the past. However, some recent studies on freeway safety could add value to the analysis of incidents.

How are the characteristic classifications of each variable in Table 1 obtained? If they are extracted from other articles, proper references should be provided.

Line 140: What is the purpose of the Spearman correlation test? What does an absolute correlation coefficient of less than 0.7 indicate? This should be explained in the text.

The conclusion of the manuscript is too extensive and repeats the content of the discussion. A brief overview of the manuscript's content and its contributions would suffice.

Reviewer #2: The manuscript proposes a latent class survival model to analyze the freeway incident duration. The topic is interesting and the manuscript is easy to follow. I have some questions/comments about the paper are as follows:

1. L27-28: Semantic relationship of some sentences in the abstract could not be established, there may be some grammar errors. Careful sentence review is very important.

2. L125: The fonts and lines provided in Figure 1 do not seem to be rigorous enough. Please review them carefully.

3. As a special freeway, the ring-road has some differences from the general freeway, which need to be explained. In addition, the existing research in the field is based on the freeway, and whether there will be significant changes in the research methods for the ring-road is worth further exploration.

4. The meaning of the Data section seems to be unclear. What is the purpose of describing the proportion of each type of incident? Does it relate to the main point or innovation of the article?

5. L156-157: There are problems with the format of the manuscript. It is very necessary to check the language and format in the manuscript. Please be more rigorous about the manuscript.

6. PLOS authors have the option to publish the peer review history of their article (what does this mean?). If published, this will include your full peer review and any attached files.

Reviewer #1: No

Reviewer #2: No

---

## [Author Response · Author response to Decision Letter 0]

24 Jun 2024

Response to the Reviewer's Comments:

Reviewer #1

Reviewer #1: Comment #1: The introduction of the article summarizes the contributions made in the field of incident duration research in the past. However, some recent studies on freeway safety could add value to the analysis of incidents.

Authors' Response: Thank you for your comments. We have discussed the latest research in depth and further combined it with this paper to add credibility and application value to this paper.

Reviewer #1: Comment #2: How are the characteristic classifications of each variable in Table 1 obtained? If they are extracted from other articles, proper references should be provided.

Authors' Response: Before analyzing the data, we first categorize it. The explanatory variables are divided into incident characteristics, temporal characteristics, environmental characteristics, traffic characteristics and operational characteristics, this classification method is quoted in other literatures, which can describe the data content more clearly and further improve the scientific nature of the paper. We have added the corresponding literature citations in "Data".

Reviewer #1: Comment #3: Line 140: What is the purpose of the Spearman correlation test? What does an absolute correlation coefficient of less than 0.7 indicate? This should be explained in the text.

Authors' Response: We have explained the purpose of Spearman correlation test in the "Data" of the revised manuscript. This is to verify that there is no multicollinearity in the optimal combination of variables after stepwise regression. At the same time, absolute correlation coefficient of less than 0.7 further proves that there is no close correlation in the combination of variables, excluding the possible influence of the additive effect of other factors.The details are as follows:

To further verify whether there is multicollinearity and tight correlation in the variable combinations after stepwise regression, the Spearman correlation test was conducted to assess the relationship between the aforementioned variables. The results revealed that the absolute values of the correlation coefficients were less than 0.7, as displayed in Table 2. This shows that there is no tight correlation in the optimal combination of variables obtained by stepwise regression, further excluding the additive effects of other factors.

Reviewer #1: Comment #4: The conclusion of the manuscript is too extensive and repeats the content of the discussion. A brief overview of the manuscript's content and its contributions would suffice.

Authors' Response: Thank you for your comments. We have further simplified the content of the "Conclusion", removing the unimportant content that repeats with the "Discussion", so as to simply describe the content and contribution of the manuscript. We believe this will increase the scientific nature of the article.

Reviewer #2

Reviewer #2: Comment #1: L27-28: Semantic relationship of some sentences in the abstract could not be established, there may be some grammar errors. Careful sentence review is very important.

Authors' Response: Semantic relationship of some sentences in the abstract has been optimized. At the same time, we have further reviewed the sentences and grammar of the paper in order to increase the credibility and scientific nature of the paper.

Reviewer #2: Comment #2: L125: The fonts and lines provided in Figure 1 do not seem to be rigorous enough. Please review them carefully.

Authors' Response: Thank you for your comment. The lines and fonts in Figure 1 were optimized to ensure that it can clearly and effectively reflect the overall idea and process of data processing.

Reviewer #2: Comment #3: As a special freeway, the ring-road has some differences from the general freeway, which need to be explained. In addition, the existing research in the field is based on the freeway, and whether there will be significant changes in the research methods for the ring-road is worth further exploration.

Authors' Response: In the penultimate paragraph of the introduction, we show the characteristics and functions of the ring-road, which are different from general freeways. In the meanwhile, it has a better supporting role for the research of this paper. Of course, because the ring road is a special freeway, there are some similarities in the research ideas. The method of this study is inspired by the previous research on the duration of freeway incidents, and the choice is further thought. Further discussion is significant on the need to find specific research methods for the ring-road. This is also one of the directions that we can study in the future. If there is a chance, we hope to continue in-depth and get guidance from experts. The specific contents are as follows:

As a special freeway in China, the ring-road is mainly used to relieve the traffic pressure in urban areas, divert vehicles, and build a three-dimensional freeway network by forming an interchange with various high-grade freeways leading to the main urban area. The ring-road has some characteristics besides the general freeway, such as weak tidal property, many entrances and exits, complex sections, etc.

Reviewer #2: Comment #4: The meaning of the Data section seems to be unclear. What is the purpose of describing the proportion of each type of incident? Does it relate to the main point or innovation of the article?

Authors' Response: Thank you for your comment. The Data section is mainly used to introduce the overall situation of the data, so that the data support of the article can be described more scientifically and clearly. At the same time, describing the proportion of each type of incident is related to the main point or innovation of the article. Most of the data introduced in the research results show that the conclusions of this paper is different from those of previous studies, which plays a role in echoing the following.

Reviewer #2: Comment #5: L156-157: There are problems with the format of the manuscript. It is very necessary to check the language and format in the manuscript. Please be more rigorous about the manuscript.

Authors' Response: We have rigorously examined the format and statements of this section and optimized them accordingly. The language and format in the manuscript also received further scrutiny.

---

## [Decision Letter · Decision Letter 1]

25 Jul 2024

Study on ring-road incident duration based on latent class accelerated hazard model

PONE-D-24-18829R1

Dear Dr. Li,

We’re pleased to inform you that your manuscript has been judged scientifically suitable for publication and will be formally accepted for publication once it meets all outstanding technical requirements.

Kind regards,

Zhihong (Arry) Yao, Ph.D.

Academic Editor

PLOS ONE

Additional Editor Comments (optional):

Reviewers' comments:

Reviewer's Responses to Questions

**Comments to the Author**

1. If the authors have adequately addressed your comments raised in a previous round of review and you feel that this manuscript is now acceptable for publication, you may indicate that here to bypass the “Comments to the Author” section, enter your conflict of interest statement in the “Confidential to Editor” section, and submit your "Accept" recommendation.

Reviewer #2: All comments have been addressed

2. Is the manuscript technically sound, and do the data support the conclusions?

Reviewer #2: Yes

3. Has the statistical analysis been performed appropriately and rigorously? 

Reviewer #2: Yes

4. Have the authors made all data underlying the findings in their manuscript fully available?

Reviewer #2: Yes

5. Is the manuscript presented in an intelligible fashion and written in standard English?

Reviewer #2: Yes

6. Review Comments to the Author

Reviewer #2: (No Response)

7. PLOS authors have the option to publish the peer review history of their article (what does this mean?). If published, this will include your full peer review and any attached files.

Reviewer #2: No

---

## [Editor Report · Acceptance letter]

1 Aug 2024

PONE-D-24-18829R1 

PLOS ONE

Dear Dr. Li, 

I'm pleased to inform you that your manuscript has been deemed suitable for publication in PLOS ONE. Congratulations! Your manuscript is now being handed over to our production team.

Kind regards, 

on behalf of

Dr. Zhihong (Arry) Yao 

Academic Editor

PLOS ONE